

# Energy efficient service placement in fog computing

Usha Vadde[1] and Vijaya Sri Kompalli[2]

[1] Department of Computer Science and Engineering, Koneru Lakshmaiah Education Foundation, Guntur, Andhra Pradash, India
[2] Department of Computer Science and Engineering, Koneru Lakshmaiah Education Foundation, Guntur, Andhra Pradesh, India

## ABSTRACT

The Internet of Things (IoT) concept evolved into a slew of applications. To satisfy the requests of these applications, using cloud computing is troublesome because of the high latency caused by the distance between IoT devices and cloud resources. Fog computing has become promising with its geographically distributed infrastructure for providing resources using fog nodes near IoT devices, thereby reducing the bandwidth and latency. A geographical distribution, heterogeneity and resource constraints of fog nodes introduce the key challenge of placing application modules/services in such a large scale infrastructure. In this work, we propose an improved version of the JAYA approach for optimal placement of modules that minimizes the energy consumption of a fog landscape. We analyzed the performance in terms of energy consumption, network usage, delays and execution time. Using iFogSim, we ran simulations and observed that our approach reduces on average 31% of the energy consumption compared to modern methods.

## INTRODUCTION

Cloud computing has gained more popularity as there is more exchange of data and information. The Internet of Things (IoT), a major tech trend, causes connecting devices to become more ubiquitous in business, government, and personal spheres. The Internet of Things (IoT) is an accolade by cloud computing with high quality caching and high-definition capabilities that enable everything to be brought online. More data is being produced by bringing in all things online (*Xu et al., 2018*), This data will be processed on the cloud. Cloud data centres are often located distant from IoT devices. This placement results in a high communication delay, but most IoT applications require low latency. The concept of fog computing enables storage, computation, and networking on the fog nodes that are closer to the IoT devices. Fog nodes can be placed anywhere between the IoT devices and the cloud path. By bringing the cloud closer to where data is created and used, fog computing with hierarchical architecture can effectively deal with latency-sensitive IoT applications. Rather than storing and processing the data on a cloud, its fog computing allows processing to be done at fog nodes in the network. In this way, information from the

Corresponding author
Usha Vadde, vuat5678@gmail.com

IoT devices can be processed separately (*Mijuskovic et al., 2021*). As the processing takes place hierarchically, total latency is decreased.

Fog computing is a novel approach with several advantages, like low cost, network bandwidth and low latency. Fog resources allow local computing and network for end-user applications. The flexibility and scalability of cloud computing will make it easier for fog computing to meet the growing need for computation-intensive and large scale applications when fog computing processing energy is insufficient.

Fog computing has several applications, including health services, surveillance, smart buildings, connected cars, and manufacturing. Fog nodes are positioned near the customer applications to keep latency and response times low. Despite advantages, there are also several challenges associated with fog computing. Managing resources properly is paramount since it will prevent downtime and energy costs.

This research focuses on one of the major fog computing challenges: module/services placement. The fog nodes are resource-constrained, so we should properly assign the modules to a fog node. Without proper allocation, the applications will starve. Proper allocation of the resources to each module can solve this issue In the literature, various optimization techniques like Ant Colony Optimization (ACO), Particle Swarm Optimization (PSO) and genetic algorithm were used. But these algorithms fall within local optimum and are sensitive to the initial population.

In the proposed algorithm, we introduced a new operator in the JAYA algorithm called Levy flight, which produces a random walk following probability distribution. We use the proposed approach for module placement in the cloud-fog environment. The Levy flight escape the locally optimal solution, resulting in an efficient placement of the modules in the fog landscape. The proposed Levy flight based JAYA(LJAYA) approach led to a fair trade-off between utilization of fog landscape and energy consumption for running applications in fog landscape.

The following are the major contributions of this research:

- Formulated service/module placement problem to minimize energy consumption.
- A new Levy flight based JAYA algorithm is proposed to solve the module/service placement problem in the fog landscape.
- Experiments for performance analysis are conducted by varying loads considering the said metrics. The results conclude that the proposed placement approach significantly optimizes the module/service placement and reduces energy consumption.

## RELATED WORKS

With the continuous development of fog computing technology, resource management has become a difficult task (*Tadakamalla & Menasce, 2021*). This section presents the existing resource management techniques with their advantages and limitations. A quick overview of some of these proposed module/service placement approaches is provided below. Fog computing deals with computationally intensive applications at the edges of the network. There exist various challenges to complex resource allocation and communication resources under QoS requirements. The issue of task scheduling and resource allocation

for multi-devices in wireless IoT networks is being investigated. *Li et al. (2019)* proposed a non-orthogonal multiple access approach. The use of various computing modes would impact the energy consumption and average delay. So the proposed method would make the optimal decision of choosing a suitable computing mode that offers good performance. The optimization issue is composed of a mixed-integer nonlinear programming problem that helps reduce energy consumption. The authors used an Improved Genetic Algorithm (IGA) to resolve this nonlinear problem. *Zhu et al. (2018)* proposed Folo, which is aimed to reduce the latency and comprehensive quality loss while also facilitating the mobility of vehicles. A bi-objective minimization problem for a task allocation to fog nodes is introduced. The vehicular network is widely adopted as a result of the imminent technologies in wireless communication, inventive manufacturing so on. *Lin et al. (2018)* investigated the resource allocation management in vehicular fog computing that aims to reduce the energy consumption of the computing nodes and enhance the execution time. A utility model is also built that follows two steps. In the beginning, all sub-optimal solutions counting on the Lagrangian algorithm are given to resolve this problem. Then, the proposed optimal solution selection procedure. QoS might get degraded for the battery-energy mobile devices due to a lack of energy supply.

*Chang et al. (2020)* proposed a technology of Energy Harvesting (EH) that helps the devices to gain energy from the environment. The authors proposed reducing the execution cost through the Lyapunov optimization algorithm. *Huang et al. (2020)* solved the energy-efficient resource allocation problem in fog computing networks. To increase the network energy efficiency, they proposed a Fog Node (FN) based resource allocation algorithm and converted it into Lyapunov optimization. Due to the immense volume of data transmissions, communication issues were increased by big data. So, fog computing has been implemented to resolve the communication issue. However, a limitation in resource management due to the amount of accessible heterogeneous computing relied on fog computing. *Gai, Qin & Zhu (2020)* addresses the problem by proposing an Energy-Aware Fog Resource Optimization (EFRO) approach. EFRO considers three components such as cloud, fog and edge layers. This approach would integrate the standardization and smart shift operations that also reduce energy consumption and scheduling length. To reduce the delays due to the inefficiency of task scheduling in fog computing, *Potu, Jatoth & Parvataneni (2021)* had proposed an Extended Particle Swarm Optimization (EPSO) that would help optimize a task scheduling problem. Load balancing techniques associated with fog computing follow two ways: dynamic load balancing and static load balancing. *Singh et al. (2020)* compared various load balancing algorithms and found a fundamentally easy round-robin load-balancing algorithm. *Jamil et al. (2020)* proposed QoS relied load balancing algorithm, the custom load method. This algorithm aims to increase the use of fog devices in a specific area while reducing energy consumption and latency. When it comes to resource optimization, linear programming is a popular approach. *Arkian, Diyanat & Pourkhalili (2017)*, in their work, suggested a mixed-integer programming approach that took into account the bottom station association as well as task distribution. *Skarlat et al. (2017)* have introduced fog colonies and used a Genetic Algorithm (GA) to decide where the services have to be placed within the colonies.

Time Cost Aware Scheduling was proposed by *Binh et al. (2018)*. The algorithm distributes jobs to the client as well as the fog layer based on overall response time, data centre costs, and processing time. However, there is no dynamic allocation of resources, and the proposed approach allocates the resources before the processing time. *Alelaiwi (2019)*, have taken this a leap forward using deep learning to optimize the response time for critical tasks in the fog landscape.

*Chen, Dong & Liang (2018)* and *Varshney, Sandhu & Gupta (2021)* focused on how a user's independent computing tasks are distributed between their end device, computing access point and a remote cloud server. To reduce the energy consumption of the above components, they employ semi-definite relaxation and a randomization mapping method. *Varshney, Sandhu & Gupta (2020)* prospered Applicant Hierarchy Processing (AHP) method for distributing applications to suitable fog layer. The suggested framework assures end-user QoE. The suggested method is evaluated for storage, CPU cycle, and processing time.

Improving the algorithm for mapping application modules/services to the fog nodes is a good research method. In the literature, module placement algorithms were proposed, but still, there is a scope for improving the optimal solution. Most of the existing solutions focused on minimizing latency in the fog landscape. This paper proposes an enhanced module placement algorithm using Levy flight. Our goal is to reduce energy consumption, network utilization and execution time.

## PROBLEM FORMULATION

The fog-cloud design takes advantage of both edge and cloud computing capabilities. Low-latency processing is carried out at lower-level fog nodes that are distributed geographically while leveraging centralized cloud services.

### Architecture of fog computing

Fog computing is a type of computing that takes place between the end node and the cloud data centre. The cloud, fog, and IoT sensors are the three layers of fog architecture. Sensors capture and emit the data but do not have the computation or storage capability. Along with sensors, we have actuators to control the system and react to the changes in the environment as detected by sensors.

Fog nodes are devices with little computing capability and network-connected devices such as smart gateway and terminal devices. This layer collect data from sensors and perform data processing before sending it to the upper layers. Fog computing is suitable for low-latency applications. As shown in Fig. 1, we extend the basic framework of fog computing in *Bonomi et al. (2014)* and *Gupta et al. (2017)* by allowing service/module placement in both the fog and cloud. For this, we introduce two levels of control: (i) cloud-fog controller and (ii) fog orchestration controller(FOC). Cloud-fog controller controls all fog nodes. Fog orchestration controllers are a special kind of fog node used to run the IoT applications without any involvement of the cloud. A fog orchestration controller is responsible for all the nodes connected to it, called a fog colony. Our fog architecture

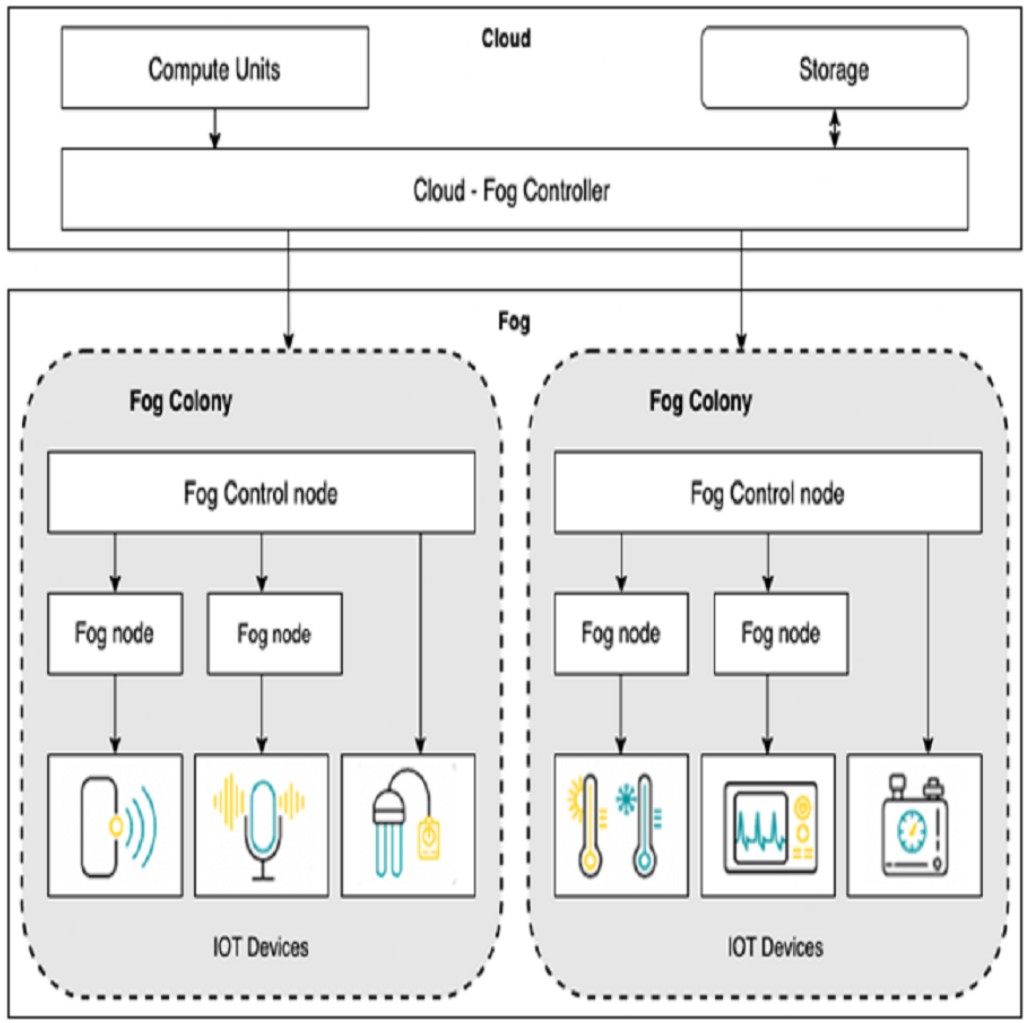

**Figure 1** Architecture for fog computing.

supports a hierarchy with the cloud-fog controller, fog orchestration controller, fog nodes, and the sensor/IoT devices at the bottom layer.

The controller nodes need to be provided with the information to analyze the IoT application and place the respective modules onto virtualized resources. For example, the fog orchestration controller is provided with complete details about its fog colony and the state of neighbourhood colonies. With this information, the scheduler develops a service placement plan and accordingly places the application modules on particular fog resources.

Fog landscape consist of set of fog nodes $(f_1, f_2, \ldots, f_n)$. These fog nodes are split into colonies, with a FOC node in charge of each colony. Each Fog node $f_j$ is equipped with sensors and actuators. Each fog node $f_j$ can be indicated with a tuple $< id, R_j, S_j, Cu_j >$ where id is the unique identifier, $R_j$ is the RAM capacity, $S_j$ is the storage capacity and $Cu_j$ is the CPU capacity of the fog node. FOC node controls all the communication within a

colony. We define a non-negligible delay $d_j$ between the FOC node and each fog node $f_j$ in that colony.

## IoT applications and services

Let $W$ denote a set of different IoT apps. The Distributed Data Flow (DDF) deployment approach is used for the IoT application, as stated in *Giang et al. (2015)*. Each of these applications $(W_k)$ is made up of several modules, where each module $m_j \in W_k$ is to be executed on the fog/cloud resources. All the modules that belong to an application $(W_k)$ need to be deployed before $W_k$ starts execution. Once the application executes, modules will communicate with each other, and data flows between modules. The application response time $r_A$ is calculated as shown in Eq. (1).

$$r_A = makespan(W_k) + deployment(W_k) \tag{1}$$

where $makespan(W_k)$ is the sum of the makespan duration of each module $m_j \in W_k$ and the execution delays. The $makespan(m_j)$ is the total time spent by the module from start to its completion. $deployment(W_k)$ is the sum of the current deployment time $deployment_{W_k}^t$ and the additional time for propagation of the module to the closest neighbour colony. We assume that the application's deployment time includes administrative tasks such as module placement. Each module $m_j$ is defined by a tuple $< CPU_{m_j}, R_{m_j}, S_{m_j}, Type >$ where these are the demands of CPU, main memory, and storage. The service type indicates specific kinds of computing resources for a module $m_j$. Our goal is to utilize the fog landscape to the maximum extent, and the placement of modules must reduce the total energy consumption of the fog landscape. This issue is referred to as Module Placement Problem (MPP) in fog landscape. The controllers monitor all the fog nodes. Each fog node $f_i$ has fixed processing power $CPU_i$ and memory $R_i$. Let $m_1, m_2, m_3, ....., m_p$ be the modules that need to be placed on to the set of fog nodes $(f_1, f_2, ......, f_n)$. This work addresses the MPP to reduce the delay in application processing and the total energy consumption of the fog landscape. A levy-based JAYA (LJAYA) algorithm for mapping modules and fog nodes has been developed. In the proposed approach, each solution is modelled by an array. This array consists of integer numbers (unique identifiers of fog nodes) corresponding to the fog node on which the modules $m_1, m_2, m_3....., m_p$ will be placed.

$Solution_i = (f_3, f_9, ....f_i, ..., f_6)$.

This solution places the $m_1$ onto $f_3$, $m_2$ onto $f_9$ etc.

## Energy consumption model

An efficient placement strategy can optimize fog resources and minimize energy consumption. Most of the previous placement algorithms have focused on enhancing the performance of the fog landscape while ignoring the energy consumption. The energy consumption by a fog node/controller can be accurately described as a linear relationship of CPU utilization (*Reddy et al., 2021*; *Beloglazov & Buyya, 2012*). We define energy consumption of a computing node $(P_i)$ considering idle energy consumption and CPU utilization (u), given in Eq. (2):

$$P_i(u) = k * P_{max} + (1 - k) * P_{max} * u \tag{2}$$

$P_{max}$ is the energy consumption of a host running with full capacity (100% utilization), k represents the percentage of power drawn by an idle host. The total energy consumption of fog landscape with n nodes can be determined using Eq. (3) (*Lee & Zomaya, 2012*).

$$E = \sum_{i=1}^{n} P_i(u). \tag{3}$$

## Module placement using Levy based JAYA algorithm

The wide spectrum of bio-inspired algorithms, emphasizing evolutionary computation & swarm intelligence, are probabilistic. An important aspect of obtaining high performance using the above algorithms depends highly on fine-tuning algorithm-specific parameters. *Rao (2016)* implemented the JAYA algorithm with few algorithm-specific parameters to tackle this disadvantage. JAYA algorithm updates each candidate using the global best and worst solutions and moves towards the best by avoiding the worst particle. This algorithm updates the solution according to Eq. (4). We have to update the population until the optimal solution is found or maximum iterations are reached.

$$Solution_{i+1} = Solution_i + r_1 * (B_i - Solution_i) - r_2 * (W_i - Solution_i) \tag{4}$$

where $Solution_i$ is the value at $i$th iteration, and $Solution_{i+1}$ is the updated value. $r_1, r_2$ are random numbers and $W_i, B_i$ are the worst and best according to the fitness value.

We modified the JAYA algorithm by introducing a new operator that searches the vicinity of each solution using a Levy flight (LF). Levy flight produces a random walk following heavy-tailed probability distribution. Levy flight steps are distributed according to Levy distribution with several small steps, and some rare steps are very long. These long jumps help the algorithm's global search capability (Exploration). Meanwhile, the small steps improve the local search capabilities (Exploitation). The updating in our approach is as follows:

$$Solution_{i+1} = Solution_i + LF(Solution_i) + r_1 * (B_i - Solution_i) - r_2 * (W_i - Solution_i) \tag{5}$$

where

$$LF(Solution_i) = 0.01 * \frac{u}{v^{1/\beta}} * (Solution_i - B_i) \tag{6}$$

where u and v are two numbers drawn from normal distributions, $B_i$ is the best solution and $0 < \beta < 2$ is an index.

Figure 2 shows the steps involved in the improved JAYA algorithm for module/service placement and are described as follows:

If the condition is true the input for next level is the updated particle, which we got after applying Eqs. (6) and (7) to the original particle. But if the condition is false then the input for next level is the original particle.

**Step 1: Initial solution**

Each solution/candidate is a randomized list where each entry specifies the fog node that satisfies the requirement of a given module. For example, the second module request will

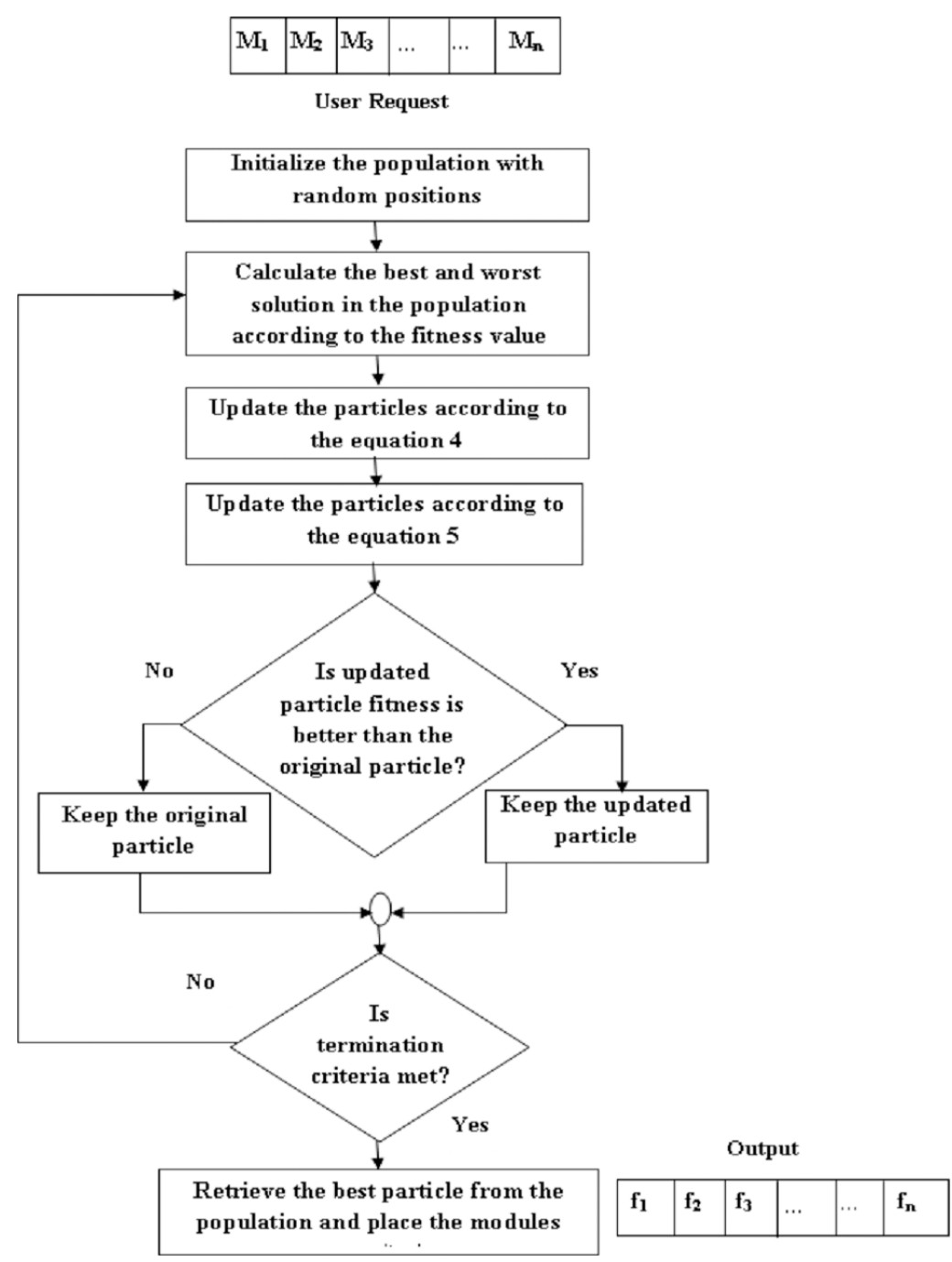

**Figure 2  Steps involved in the proposed algorithm.**

be placed on the fog node given as the second element of the list. Then the fitness for each solution is calculated as shown in Eq. (3).

**Step 2: Updation**

Calculate the fitness of each candidate and select the solutions that lead to higher and lower fitness (energy consumption in our case) values as the worst and best candidates. The movement of all the candidates is revised using the global best and worst according to Eq. (4). This equation changes the candidate's direction to move towards better solution areas.

**Step 3: Spatial dispersion**

To improve the exploration and exploitation of the particles we add the Levy distribution to the updated particles, as shown in Eq. (5). We keep $Solution_{i+1}$, if it is the promising solution than the $Solution_i$. In the next iteration, we apply these operations to the updated population. During this process, all candidates move towards optimal solutions keeping away from the worst candidate.

**Step 4: Final selection**

All the particles are updated until the global optimum is found or the number of iterations is over. Finally, the solution with the highest fitness value is selected, and modules are placed on the respective fog nodes.

## PERFORMANCE EVALUATION

We simulated a cloud-fog environment using iFogSim (*Gupta et al., 2017*). It is a generalized and expandable system for simulating various fog components and real time applications. iFogSim allows simulation and the evaluation of algorithms for resource management on fog landscape. iFogSim has been used by most universities and industries to evaluate resource allocation algorithms and for energy-efficient management of computing resources. So, we also used the iFogSim to simulate our experiments. We analyzed the proposed approach concerning energy consumption, delays, execution time, network usage, etc. We have considered Intelligent Surveillance through Distributed Camera Networks (ISDCN) for our work. Smart camera-based distributed video surveillance has gained popularity as it has lot a of applications like linked cars, security, smart grids, and healthcare. However, multi-site video monitoring manually makes the surveillance quite complex. Hence we need video management software to analyze the feed from the camera and provide a complex solution such as object detection and tracking. Low-latency connectivity, handling large amounts of data, and extensive long-term processing are all required for such a system (*Gupta et al., 2017*).

When motion is detected in the smart camera's Fields Of View (FOV), it begins delivering a video feed to the ISDCN application. The target object is identified by the application and located in its position in each frame. Moving object tracking is accomplished by adjusting camera parameters from time to time. ISDCN application comprises five modules, as shown in Fig. 3. The first module is Object Detector which identifies an object in a given frame. The second module is for Motion Detection, and the third module tracks the identified object over time by updating the pan-tilt-zoom (PTZ) control parameters. The user interface is to display the detected object. A detailed description of these modules is

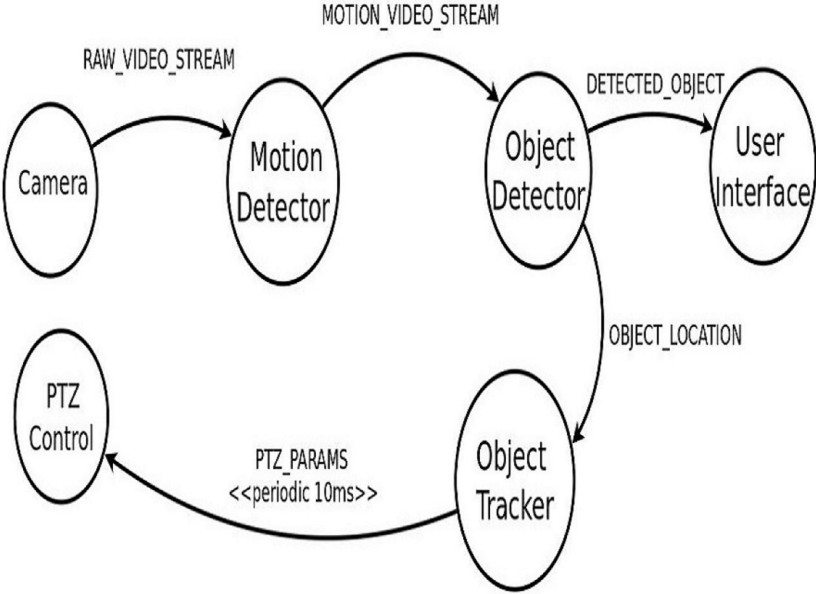

**Figure 3** **Modelling of the ISDCN application.**

given in *Gupta et al. (2017)*. The application will take the feed from the number of CCTV cameras, and after processing these streams, the PTZ control parameters are adjusted to track the object. The edges connect the modules in the application and these edges carry tuples. Table 1 lists the properties of these tuples.

Table 2 shows the different types of fog devices employed in the topology and their configurations. Here, the cameras serve as sensors and provide input data to the application. On average, the sensors have 5-millisecond inter-arrival times, which require 1000 MIPS and a bandwidth of 20,000 bytes. The physical topology is modelled in iFogSim using the FogDevice, Sensor and Actuator classes.

## Results and Discussion

This section presents the results of the proposed module placement algorithm for the ISDCN application and compares them with state-of-the-art approaches in terms of energy, latency, and network utilization. We compared the proposed module placement approach with the approaches like EPSO (*Potu, Jatoth & Parvataneni, 2021*), PSO (*Mseddi et al., 2019*), JAYA (*Rao, 2016*), and Cloud Only (*Gupta et al., 2017*). To compare the performance of these approaches, we perform several experiments using the same physical topology of the ISDCN application and varying the number of areas.

The proposed approach is evaluated on ISDCN application by varying the number of areas with four cameras. All the cameras are connected to the cloud *via* a router in a cloud-only approach.

**Table 1 Details of the edges in the ISDCN application.**

| Tuple type | MIPS | Network bandwidth |
|---|---|---|
| OBJECT LOCATION | 1000 | 100 |
| RAW VIDEO STREAM | 1000 | 20000 |
| PTZ PARAMS | 100 | 100 |
| MOTION DETECTION | 2000 | 2000 |
| DETECTED OBJECT | 500 | 2000 |

**Table 2 Characteristics of the Fog devices used for ISDCN.**

| | CPU MIPS | RAM (MB) | Uplink Bw (MB) | Downlink Bw (MB) | Level | Rate per MIPS | Busy power (Watt) | Idle power (Watt) |
|---|---|---|---|---|---|---|---|---|
| Cloud | 44800 | 40000 | 100 | 10000 | 0 | 0.01 | 16*103 | 16*83.25 |
| Proxy | 2800 | 4000 | 10000 | 10000 | 1 | 0 | 107.3 | 83.43 |
| Fog | 2800 | 4000 | 10000 | 10000 | 2 | 0 | 107.3 | 83.43 |

*Energy consumption analysis*

Figure 4 shows the superior performance of the proposed LJAYA algorithm in terms of the energy consumption for all the configurations measured in Kilo Joules (kJ) A lot of energy is consumed by the cameras to detect the objects' motion in frames. Total energy consumption was significantly less in the LJAYA method than in JAYA, EPSO, PSO, and Cloud Only. For instance, the total energy consumption with EPSO, JAYA, PSO and Cloud Only is 509.12 kJ, 523.39 kJ, 689.48 kJ, and 1915.10 kJ. In comparison, the LJAYA method was 480.10 kJ for ten areas. When the number of areas is increased, the total energy consumption also increases with all the approaches. The proposed approach can find the optimal solution in all the cases. The analysis of the energy consumption for various configurations demonstrated that the proposed LJAYA approach reduces energy consumption up to 31% on average compared to modern methods.

*Execution time analysis*

Figure 5 shows the execution time (in milliseconds) of various topologies and input workloads. From Fig. 5, it is clear that the proposed LJAYA approach can complete the execution faster than the other approaches. On average, the proposed approach reduced the execution time up to 7%, 15%, 22%, and 53% over EPSO, JAYA, PSO, and Cloud Only approach, respectively.

*Network usage analysis*

The network usage will increase if traffic is increased toward the cloud. At the same time, the network usage decreases when we have a dedicated fog node in each area. The network usage is calculated using Eq. (7) (*Gupta et al., 2017*).

$$Networkusage = Latency * \delta, \tag{7}$$

where $\delta$ = tupleNWSize.

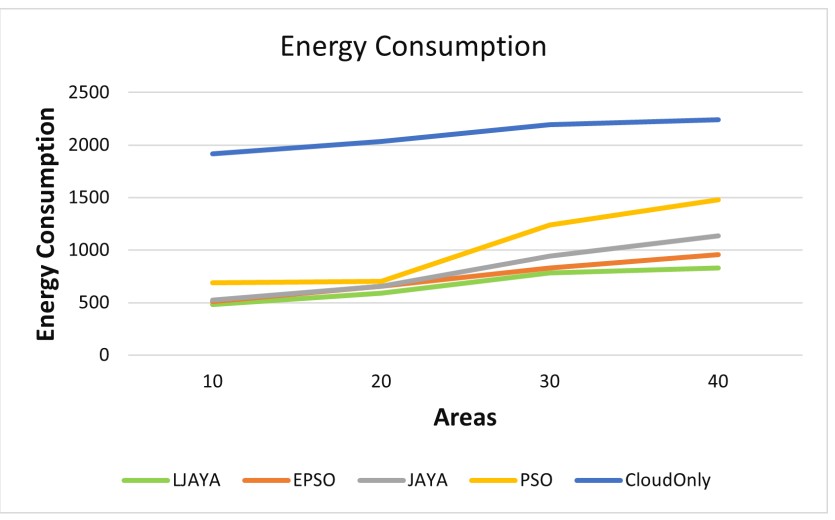

**Figure 4** Energy consumption of all devices in fog landscape.

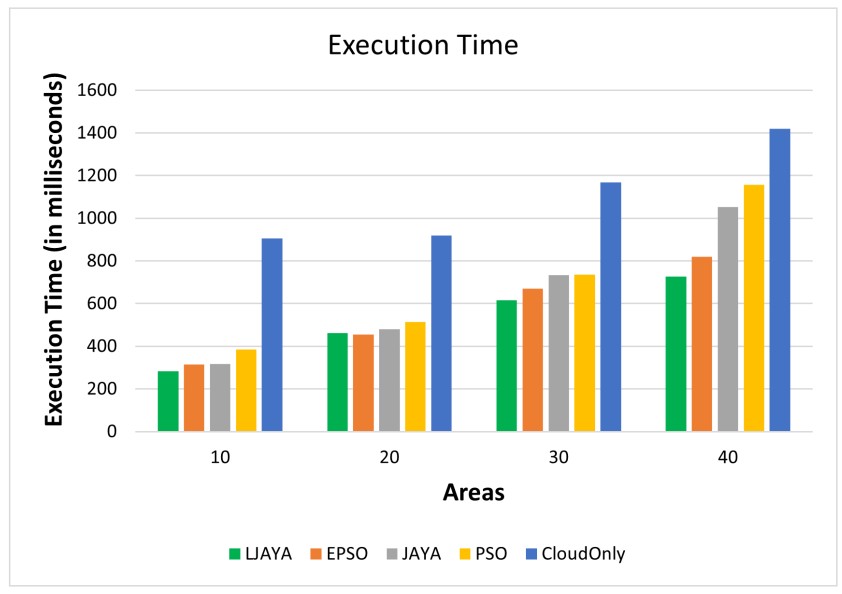

**Figure 5** Execution time analysis.

Experimental results in terms of the network usage in bytes are shown in Table 3. Network usage is high with the cloud-only approach because all processing happens in a cloud server. But, with the proposed approach, processing occurs at efficient fog nodes, reducing the network usage. Considering 40 areas, the network usage with the proposed LJAYA, EPSO, JAYA, PSO, and CloudOnly are 2,483,404 bytes, 2,485,275 bytes, 2,485,814 bytes, 2,487,663 bytes, and 2,991,055, respectively. We can reduce the network usage by up to 16% using the proposed approach when compared to the CloudOnly approach.

**Table 3  Total network usage in bytes.**

| Areas | LJAYA | EPSO | JAYA | PSO | CloudOnly |
|---|---|---|---|---|---|
| 10 | 1466620 | 1466806 | 1466804 | 1467504 | 1474585 |
| 20 | 1972125 | 1972196 | 1972271 | 1974304 | 1980075 |
| 30 | 2478204 | 2480074 | 2482234 | 2482234 | 2485565 |
| 40 | 2483404 | 2485275 | 2485814 | 2487663 | 2991055 |

**Table 4  Latency analysis in ms.**

| Areas | LJAYA | EPSO | JAYA | PSO | CloudOnly |
|---|---|---|---|---|---|
| 10 | 1.1 | 2.2 | 2.2 | 20.899 | 105.999 |
| 20 | 2.16 | 3.3 | 4.3 | 30.9 | 105.999 |
| 30 | 2.89 | 3.3 | 7.015 | 31.7 | 105.999 |
| 40 | 3.2 | 5.4 | 19.9 | 32.6 | 105.999 |

### Latency analysis

Real-time IoT applications need high performance and can achieve this only by reducing latency. The latency is computed using Eq. (8) (*Gupta et al., 2017*).

$$Latency = \alpha + \mu + \theta \tag{8}$$

where $\alpha$ is the delay incurred while capturing video streams in the form of tuples and $\mu$ is the time to upload and perform motion detection. Finally, $\theta$ is the time to display the detected object on the user interface.

Experimental results in terms of latency are showed in Table 4. All application modules are placed in the cloud in a cloud-only placement algorithm, causing a bottleneck in application execution. This bottleneck causes a significant increase (106 ms) in the latency. On the other hand, the proposed placement approach can maintain low latency (1.1 ms) as it places the modules close to the network edge. Compared with the other algorithms, the proposed LJAYA approach shows superior performance in minimizing execution time, latency and energy consumption.

## CONCLUSION

Cloud and fog computing oversee a model that can offer a solution for IoT applications that are sensitive to delay. Fog nodes are typically used to store and process data near the end devices, which helps to reduce latency and communication costs. This article aims to provide an evaluation framework that minimizes energy consumption by optimally pacing the modules in a fog landscape. An improved nature-inspired algorithm LJAYA was used with levy flight and evaluated the performance in various scenarios. Experimental results demonstrated that the LJAYA algorithm outperforms the other four algorithms by escaping from the local optimal solutions using levy flight. With the proposed algorithm, we can reduce the energy consumption on average by up to 31% and execution time up to 53%. In the future, we plan to consider different applications and propose an efficient resource provisioning technique by considering the application requirements.

### Funding

The authors received no funding for this work.

### Competing Interests

The authors declare there are no competing interests.

### Author Contributions

- Usha Vadde conceived and designed the experiments, performed the experiments, analyzed the data, performed the computation work, prepared figures and/or tables, authored or reviewed drafts of the article, and approved the final draft.
- Vijaya Sri Kompalli conceived and designed the experiments, authored or reviewed drafts of the article, and approved the final draft.

### Data Availability

The code is available in the Supplementary File.

### Supplemental Information

Supplemental information for this article can be found online at http://dx.doi.org/10.7717/peerj-cs.1035#supplemental-information.

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
