# Peer review of "Energy efficient service placement in fog computing"

_PeerJ Computer Science, doi:10.7717/peerj-cs.1035_

## Round 0.1 · original submission · Major Revisions

The reviewers raised some concerns that need to be clarified in the revised version. Please also note that I do not expect you to cite any references recommended by the reviewers unless they are critical. Please provide a detailed response letter. Thanks.

Reviewer 1 ·

Basic reporting

1. Authors are required to improve the English language and some sentence structures need to be updated.
2. Some cited references are very old of year 2012
3. Overall article structure is good.
4. Hypotheses and results are OK and shared with the RAW data.
5. In some sections authors have written too many paragraphs for expressing their views. Need to update the style with some professional writing style.
6. Provide expanded from of abbreviations wherever used for the first time.

Experimental design

Here, authors have proposed an improved version of JAYA approach for optimal placement of modules
that minimizes the energy consumption of a fog landscape. They have analyzed the performance in terms of energy consumption, network usage, delays and execution time.

- Authors need to provide the details about the JAYA approach. As authors have cited that this approach is published @ref[19] but this publication is not a standard one and there is a doubt that the journal is a predatory journal. Authors are required to update this citation with some published/ cited with some standard publisher.

- Make your Introduction section more technical than in the current form it just focuses on the basics of fog computing.

- What is the worst-case time complexity of the modified JAYA approach and previously published JAYA approach.

- In the literature survey, authors are discussing vehicular communication.... why??

- Kindly refer the following work for further improvement in the literature survey section
10.1109/ISPCC53510.2021.9609479
10.4018/IJKSS.2020100102


- Why authors have not applied any MCDM based approach for the allocation of resources and services in the Fog landscape?

- Equation(3) is a general equation and computes the energy consumption. But how the Eq 2 has been derived?
- In the figure 2, there is a flaw in the flowchart. As the output from the first decision box irrespective of the condition whether it is true or false is becoming the input for the next level.....Why??

- What is the purpose of Figure 3?/

- Figure 4 doesn't include the unit of energy consumption.

- Provide the latency comparison of all the approaches.

Validity of the findings

Obtained results are valid.

Additional comments

NIL

Reviewer 2 ·

Basic reporting

The paper is generally well-written and reasonably structured.

Experimental design

The authors have explained their system model and the proposed algorithm algorithm very well. However, they need to further clarify some points such as:
- The complexity of the proposed algorithms are not analysed.
- Some details about where (on edge-devices, on gateways, in cloud?) the proposed algorithms are run are also needed. Also, it is unclear how these algorithms are triggered to run. The algorithms are run every time a new application appears? Some details are needed;

Validity of the findings

The authors have simulated several representative scenarios. However, they need to compare their results to other relevant examples such as the Edgeward algorithm inherently built within the iFogSim simulator. In addition, they are kindly requested to compare their work to other closely related efforts such as the fowling papers:

https://dl.acm.org/doi/10.1145/3344341.3368795

https://link.springer.com/article/10.1007/s12652-021-02910-w

- For the results shown in Fig. 4, the authors are requested to show the percent decrease in energy dissipation as compared to each existing algorithm. In addition, it can be notices that the EPSO algorithm has achieved very close numbers to the proposed algorithm. In other words, they have mentioned that they have achieved 31% average improvement considering the Cloud-only algorithm which is considered as a naïve benchmark in presence of a bulk of recently proposed application placement algorithms. Moreover, they have less that 3% improvement as compared to the EPSO algorithm. In addition, they need to compare their results to other closely related efforts such as the aforementioned two papers.
- Overall, the performance of the proposed algorithms is relatively close to that of the EPSO algorithm. Hence, the authors need to show what significant advantages their proposed algorithm can achieve.

---

## Round 0.2 · accepted · Accept

The paper can be accepted. Congratulations.

Reviewer 1 ·

Basic reporting

Authors have improved the language.
References have been updated that includes some more recently published work
Figures are OK.

Experimental design

The authors have carried out the various experiments as per identified problem statement.
They have updated the methodology section accordingly.

Validity of the findings

The proposed solution is novel and supported by results and comparative studies with previous approaches.